# The Blue Swimming Crab *Portunus segnis* in the Mediterranean Sea: Invasion Paths, Impacts and Management Measures

**DOI:** 10.3390/biology11101473

**Published:** 2022-10-08

**Authors:** Luca Castriota, Manuela Falautano, Teresa Maggio, Patrizia Perzia

**Affiliations:** Italian Institute for Environmental Protection and Research, Department for the Monitoring and Protection of the Environment and for the Conservation of Biodiversity, Unit for Conservation Management and Sustainable Use of Fish and Marine Resources, Lungomare Cristoforo Colombo 4521 (Ex Complesso Roosevelt), Località Addaura, 90149 Palermo, Italy

**Keywords:** alien species, biological invasions, ecological indicators, ecosystem services, GIS, invasion dynamics, invasion hotspots, mitigation measures, spatio–temporal statistics, systematic review

## Abstract

**Simple Summary:**

The blue swimming crab *Portunus segnis* is native to the western Indian Ocean, including the Red Sea and the Persian Gulf. It entered the Mediterranean Sea through the Suez Canal at the end of the 19th century and has colonized most of the eastern Mediterranean coasts, becoming an invasive species. This study describes the distribution, aggregation patterns, and spatial structure of this species in the Mediterranean Sea through a series of ecological indicators. The results indicated a long phase of establishment and a recent, rapid expansion phase. Several settlement areas, where the species continues to be reported, were found. Since 2015 *P. segnis* has shown a conspicuous presence in Tunisia from where it is spreading northward and eastward. The study provides an insight on the impact of *P. segnis* on biodiversity and ecosystem services and proposes a series of desirable management actions to mitigate the expansion of its population.

**Abstract:**

Invasive alien species represent one of the main environmental emergencies and are considered by the scientific community as being among the leading causes of biodiversity loss on a global scale. Therefore, detecting their pathways, hotspot areas and invasion trends becomes extremely important also for management purposes. A systematic review on presence of *Portunus segnis* in the Suez Canal and Mediterranean Sea was carried out in order to study the invasion paths from its entry from the Red Sea into the Suez Canal (1886) until recently (2021) through ecological indicators elaborated with GIS spatial–temporal statistics. Arrival, establishment and expansion phases and areas of *P. segnis* in the Mediterranean were identified. Settlement areas were detected along the Suez Canal as well as in the Levantine Sea, western Ionian Sea and Tunisian plateau ecoregions. Since 2015 a persistent area has formed in Tunisia from where the species is spreading northward and eastward. The study provides an insight on the impact of *P. segnis* on biodiversity and ecosystem services and proposes a series of desirable management actions to mitigate the expansion of its population. Following the 8Rs model that introduces the rules to mitigate non–indigenous species pollution, six of them (Recognize, Reduce, Replace, Reuse, Remove, and Regulate) have been identified as applicable and are discussed.

## 1. Introduction

The study of alien species, and in particular of invasive species (IAS), is certainly among the most debated issues of recent decades, both for the dramatic growth of the phenomenon everywhere and for the implications it has on the environment, health and economy on a local and global scale [1]. Despite the increasing number of scientific studies focusing on several aspects of the subject, from the detection and monitoring of IAS to their management, there are still many topics that have not yet been clarified. For example, the range of natural origin of many aquatic species is not always certain due to gaps in scientific knowledge and missing or controversial data on their distribution, so that they cannot clearly be ascribed as native or non–indigenous. These species are referred to as cryptogenic [2] and, according to recent estimates, amount to 440 in Europe and its neighbouring regions [3]. 

In many cases, the vectors of introduced species are not known and the invasion paths are not always fairly traceable, although assumptions can be made based on spatial statistical studies and molecular analyses as well. Exemplary is the case of *Bursatella leachii* Blainville, 1817, a pantropical sea slug that was recorded for the first time in the Mediterranean presumably around the 1940s and was initially thought to have entered the Mediterranean from the Red Sea [4]. Molecular analyses have suggested a natural immigration from the tropical Atlantic through the Strait of Gibraltar [5], but some scientists are still leaning towards the Lessepsian immigration hypothesis [6] and still the case remains unsolved. Sometimes it is extremely difficult to reconstruct the pathway of a species when, for example, it is transferred from one place to another through anthropic vectors and therefore its presence is recorded in different locations without any apparent logical route. This might be the case of the Atlantic moray *Enchelycore anatina* (Lowe, 1838), considered a cryptogenic species in the Mediterranean Sea; evidences of an established population and settlement areas in the central Mediterranean and Levant Sea have been confirmed by the spatial statistics, supporting the hypothesis of human–mediated introduction [7]. 

It is therefore important to encourage studies that can help solve these questions, providing an overview of IAS spatial–temporal distribution and identifying directional trends of spread. One method of such studies that allows us to understand the settlement areas and/or diffusion phases of non–indigenous species (NIS) is based on the analysis of scientific literature records, eventually integrated with reports from citizen science activities [8]. In fact, some of these studies have already proved useful in identifying new hotspot areas and interpreting the spatial–temporal distribution of IAS in recently colonized environments [9], suggesting alternative hypotheses about their introduction in the Mediterranean compared to those more accredited in the most recent scientific literature [7]. 

Considering the importance of the phenomenon of IAS, understanding their pathways, hotspot areas and invasion trends becomes extremely important also for management purposes. Among the alien species that have invaded the Mediterranean Sea, the portunid crab *Portunus segnis* (Forskål, 1775), commonly referred to as the blue swimming crab or blue crab, draws the attention of researchers and decision makers, on the one hand because of its potential impact on biodiversity, being a large omnivorous predator [10,11], and on the other because it could represent a valuable fishery resource due to its flavourful meat. The species is native to the western Indian Ocean, including Red Sea and Persian Gulf [12]. It is a littoral species with nocturnal habits, living in several types of substrates (rocky, muddy, sandy) in shallow waters (intertidal to 55 m), occasionally found in estuaries and in a wide range of inshore and continental shelf areas including algal or seagrass habitats. It can be considered an opportunistic predator and its diet is primarily carnivore [13,14]. The breeding season extends throughout the year except winter, and spawning is related to seasonal changes of sea surface temperature and can occur in different periods of the year [10,15]. According to these life history characteristics *P. segnis* can be classified as an invasive species. It is one of the earliest introductions through the Suez Canal, having been recorded in Port Said, Egypt, in 1898 [16]. In the present paper, we study the invasion paths of *P. segnis* in the Mediterranean Sea from the moment of its entry into the Suez Canal from the Red Sea to the most recent times, through the use of spatio–temporal ecological indicators, according to Perzia et al. [7], with an insight on impacts and management measures.

## 2. Materials and Methods

### 2.1. Systematic Review of *P. segnis* Invasion History in Mediterranean Sea 

A systematic review was carried out to collect all the available information and data on *Portunus segnis* necessary to describe the distribution, aggregation patterns, and spatial structure of this species in the Mediterranean Sea and Suez Canal. The systematic review was carried out according to the orientations of Preferred Reporting Items for Systematic Reviews and Meta-Analyses (PRISMA) [17]. Literature from the ISPRA Database of marine alien species in the Mediterranean Sea, used to build the atlas “Atlante delle specie non indigene nei mari italiani e nel Mediterraneo” (updated to 2011) [18] was a valid starting point for the review. This reference list was then updated with other studies identified using Google Scholar (scholar.google.com) freely accessible web search engine of academic articles (accessed up to December 2021), as well as consulting the online archives of the main scientific journals that publish studies on bioinvasion topics. 

Several combinations of keywords were used to identify relevant literature: “*Portunus segnis*”, “*Portunus pelagicus*”, “*Neptunus pelagicus*”, “*Lupa pelagica*”, “Blue crab”, “Blue swimming crab”, “Mediterranean”, “Suez”.

The reference lists of the publications found were also used as bibliographic sources. 

Only the studies containing geographical references of the records of *P. segnis* (i.e.: exact coordinates, map, city name, gulf name, sea name) were included in the review. Citizen science archives (e.g., iNaturalist.com) were also consulted. However, following a recent revision of the *Portunus pelagicus* species complex based on molecular and morphological analyses which recognise four individual species, also including *P. segnis* [12], we decided not to include the reports from those archives because the validation of the species is not always certain. Literature reporting only generic, doubtful or duplicate information was excluded. 

Information and data relevant for this study were extracted and organised into a database. The data warehouse included year of record (if missing, the year of publication was used), location, country, specimens’ abundance, sex, measures and weight of the individuals, habitat, depth, method of collection/observation and bibliographic source. 

All data were structured on a geographical reference and an accuracy value has been ascribed to the geographic coordinates: 1. exact coordinates or detailed map reported on the document; 2. indication of the specific sighting/capture site (e.g. city name); 3. indication of the generic sighting/capture site (e.g. gulf name); 4. highly generic location (e.g. sea name). 

The cumulative curve of *P. segnis* occurrences was calculated and divided into arbitrary intervals based on the most evident slope changes, corresponding to the phases of invasion [7,19]; for each time interval identified, the equation of the regression line was calculated in order to obtain the different rate of occurrences increase over time. 

### 2.2. Distribution, Aggregation Patterns, and Spatial Structure Analyses

The spatial data and their attributes were processed under the Geographic Information Systems (GIS) using ArcGIS 10.3 ESRI considering only the year of first record within a 0.05° Lat/Long grid, and not the number of specimens, to reduce the effect of possible preferential sampling [20]. 

The quantitative multi–parameter modelling of the dataset was carried out through the ‘Density’ toolset in ArcGIS spatial analysis toolbox and ‘Analysing patterns’, ‘Mapping clusters’ and ‘Measuring geographic distributions’ into ArcGIS spatial statistics toolbox, in order to describe the distribution, aggregation patterns, and spatial structure of *P. segnis* occurrences in the Mediterranean Sea [21,22,23].

According to Perzia et al. [7], a series of ecological indicators were examined in order to study: the temporal pattern through the population’s increasing rate over time based on evident slope changes of the cumulative curve of all occurrences;the spatio–temporal pattern through cumulative kernel density of occurrences in order to investigate the population’s increasing rate over time and space, and the evolution of aggregation nuclei into persistent areas;the aggregation patterns and spatial structure via the spatial pattern at global scale (dispersion vs. random vs. cluster) and at local scale (hot spot, cold spot, clusters and outlier), in order to highlight the change over time, the direction of spread, the dispersion/settlement areas and outliers;the key characteristics of distribution (centre of gravity, directional dispersion and directional trends) by tracking changes in shape distribution (dispersed, compact, or elongated) over time and space and comparing the time group of occurrences with each other.

In Table 1 the set of analyses and indicators used in this study is reported, including the tools, the spatial and time scale and the ecological meaning. 

## 3. Results

The literature search approach resulted in 3757 studies. After the screening (removal of duplicates and non-informative studies), 93 studies published from 1888 to December 2021 were assessed for eligibility and considered for review (Figure 1). They included 281 records of *Portunus segnis* distributed in twelve countries. 

### 3.1. Invasion History and Spatial–Temporal Patterns of *P. segnis* Distribution in the Mediterranean Sea

The cumulative curve resulting from the occurrences of *P. segnis* in the Suez Canal and in the Mediterranean shows three phases in the invasion process, here indicated as: (i) arrival from 1886 to 1923, (ii) establishment from 1924 to 2001, and (iii) expansion from 2002 to 2021 (Figure 2). The first signs of invasion of *P. segnis* from the Red Sea towards the Mediterranean Sea are between 1886 and 1893 when the first sightings along the Suez Canal were recorded [16,24], and the first observation in the Mediterranean Sea occurred in Port Said in 1898 [16]. 

The slope-positive variations (i.e., increasing rate in occurrences over time) indicated that the *P. segnis* population in the Mediterranean has grown much faster in the last twenty years (with a slope of 8.79 ± 0.56, expansion phase) than during the previous one hundred years (arrival and establishment phase with slope of 0.13 ± 0.01 and 0.91 ± 0.02 respectively), with an acceleration phase in the last six years. 

The analysis of the cumulative curve and Kernel density maps (Figure 3a–i) showed a plateau phase of about 25 years (arrival phase), during which a very low number of records have been recorded in the Suez Canal and Mediterranean Sea (four and two records respectively) (Figure 3b). This period was followed by a very long establishment phase of about 78 years (from 1923 to 2001), during which there was a slow increase in the number of records. The crab simultaneously colonised the coasts of Israel and Egypt [16] and then expanded northwards until it reached Turkey in 1928 [25] (Figure 3c). In addition to the findings in Cyprus [26] and south–western Turkey [27], the subsequent spread of *P. segnis* was observed only in the early 1960s when the species was found on several occasions in eastern Sicily (Italy) [28,29,30]. The population of *P. segnis* in the eastern sector of the Mediterranean intensified, and in the central sector two specimens were found in Malta, misidentified with *Callinectes sapidus* [31], and in the western basin the species was found at the larval stage in the Gulf of Tigullio (north–western Italy) [32] (Figure 3d). In the 90s, the species was recorded for the first time in Greece in a low number of specimens [33] (Figure 3e).

The Kernel density maps (Figure 3) show substantial changes over time and space and highest density areas (occurrence persistent areas) are evident. A first nucleus of aggregation (blue area in Figure 3f–g) along the Levantine coasts was found, representing the initial spread area of the *P. segnis* Mediterranean population; this nucleus was strengthened over time by other occurrences in the immediate neighbouring areas, from Egypt to Israel and in the Suez Canal. A second evident aggregation centre appeared along the Mediterranean Turkish coast, on the border with Syria (Figure 3h). Only since 2014, it has been colonising Tunisian coasts. This third nucleus of aggregation in Tunisia, together with that between Sicily and Malta (Figure 3i), assumed particular importance in the last six years, where *P. segnis* is still present with a permanently settled population [34,35].

### 3.2. Aggregation Patterns and Spatial Structure 

The distribution of *P. segnis* in the Mediterranean Sea shows a weak spatial autocorrelation at global scale (GMI = 0.32; z > 2.58; *p* < 0.01), indicating a change in the spatial pattern over time. 

Figure 4 provides an overview of the clusters and outliers analyses (AMI) and hotspots (GOG*). At local scale, the coasts of the eastern basin show a statistically significant clustering of low values and a cold spot on the Egyptian–Israeli coast, and another low-values cluster in Turkey (near Syria), corresponding to the older records of *P. segnis* in the Mediterranean Sea. A high value cluster and a hot spot were found in Tunisia, corresponding to the most recent records. Outliers of high value were found in the Suez Canal and in Egypt. The other records have non–significant GOG* and AMI values.

### 3.3. Key Characteristics of Distribution

Table 2 provides the key characteristics of *P. segnis* distribution in the Mediterranean Sea: central tendency (mean and median centre), directional dispersion and directional trend, calculated for 1886–1937, 1950–1978 and 1987–2021 periods.

The values show that the distribution changes in space and time. From 1886, i.e., the year of the first record in the Suez Canal, to 1937 the central tendencies, measured as median and mean centre, were found in the Canal and just outside in the Mediterranean Sea respectively, both median and mean centre were in close proximity. The directional dispersion of distribution and trends were concentrated along the coast, from Egypt to Lebanon (Figure 5).

In the 1950–1978 period the median centre was located in the centre of Levant, whereas the mean centre was near the coast of Turkey and Greece. The spatial dispersion showed a considerable change in shape and direction, showing a high westward dispersion.

Such expansion is also confirmed by the distribution key characteristics in the third period (1987–2021) that quantitatively showed the higher dispersion of *P. segnis* over time. The ellipse is elongated from east to west and the direction extends towards the centre of the Mediterranean Sea (Strait of Sicily). Central tendencies are shifted further west.

## 4. Discussion

### 4.1. *P. segnis* Invasion Pathways

The blue swimming crab *Portunus segnis*, previously referred to as *Lupa pelagica*, *Neptunus pelagicus* or *Portunus pelagicus*, was the first Lessepsian immigrant crustacean to be noticed in the Mediterranean after the opening of the Suez Canal [36]. The passage of the crab along the 162.5 km route from Port Taufiq in the Gulf of Suez (Red Sea) to Port Said in the Mediterranean, could have taken place when the dilution of the hyperhaline Bitter Lakes by the currents from the Red Sea began. Until 1869, in fact, the high salinity recorded both at the surface and at the bottom (50–52‰ and 68–80‰ respectively) of the Great Bitter Lake [37] represented a barrier for the passage of organisms. The first record of *P. segnis* in the Bitter Lakes is instead in 1886 [24], most likely when the salinity there had fallen below the prohibitive threshold. A study carried out on *P. pelagicus sensu lato* indicated that a salinity range outside 20–35 ppt can significantly reduce survival, growth, and development of early juvenile blue swimmer crabs [38]. Currently, *P. segnis* is present massively and throughout the year with established populations along the Suez Canal, being the most represented brachyuran crab species [39], as also indicated by Anselin local Moran’s I in the present study (Figure 4). Once the Mediterranean was reached, about 20 years passed before the crab began to settle permanently, a period corresponding to the arrival phase. In this relatively long period the species appeared to be confined to that area (Figure 3b) and it did not show any population growth, indicating a distinct lag phase as typical of biological invasions according to [40]. After this period, *P. segnis* began to increase its population on the east coast of the Mediterranean Sea so much so that in the 1920s it was sold in the fish markets of Haifa and Alexandria [16] as well as in Siria–Turchia [25]. In the following years, the species also invaded the western Ionian Sea. This period corresponds to the establishment phase which lasted about 80 years. The discontinuity of *P. segnis* records between the Levant and the Ionian Sea suggests human-mediated transfer probably of larval stages from the former to the latter area, as also suggested by Ariani and Serra [28]. The expansion phase began after 2001 during which the species first strengthened the Levantine population and successively the Ionian one. Between 2010 and 2015 a further spread of the species was observed from Sicily–Malta towards the eastern Tunisian coasts, which rapidly became a hotspot and a persistent area (as indicated by GOG* z–score and the high kernel density values in this area). Its success in this area has been attributed to not having predators and not being a target species of local markets [41]. However, the former hypothesis seems unrealistic since blue crabs generally have several predators at different stages of their life: eggs can be predated by fish, larval stages can be eaten by several planktivore organisms, andjuveniles and adults can be preyed upon by large fish, turtles, seabirds and mammals, as happens for the morphologically similar confamilial *Callinectes sapidus* [42]. Also, for the latter hypothesis, according to [43] blue swimming crab has become frequent in all Tunisian fish markets and in high demand from consumers, leading to the development of fisheries addressed to catch this species. Likely, other factors also come into play, not least the increasing anthropic pressure on the littoral zone that would weaken native communities, as well as climate change that would favour the settlement of tropical and subtropical species at the expense of native ones [44]. After 2015, from the Tunisian eastern coasts the invasion of *P. segnis* proceeded in two opposite directions—towards northern Tunisia on one side and towards Libya on the other side where the species has been recorded in 2021 [45]. Also, the Greek–Turkish population tended to increase and weak expansion signs from Egypt to eastern Libya started to appear in this period. 

Overall, according to GOG* z–scores non–significant values (Figure 4), settlement areas where *P. segnis* is constantly present over time have been identified in Lebanon, Syria, Turkey and Cyprus as well as in Sicily–Malta. However, sectors with not significant values turned out to be present also along the Libya–Egypt coasts and in the western Mediterranean, but they were dispersed in space and time so that they should be considered as casual occurrences.

Despite the several features of invasiveness shown by this species, such as continuous spawning periods, high fecundity, rapid growth, omnivorous habits and opportunistic feeding behaviour [46,47,48,49], overall, its invasion in the Mediterranean Sea seems to proceed very slowly. There are even some areas, e.g., the Aegean Sea, where *P. segnis* has been recorded relatively rarely and only in the southern area, which seems to be resistant to invasion although this species is able to live in a wide variety of habitats. In the Arabian Gulf it inhabits mainly coastal waters in seagrass beds and mangrove [11], but elsewhere it has also been found under rocks and in rock pools, on sandy or muddy substrate [13] as well as in lagoon environments [43]. However, colder temperatures in the north Aegean Sea than in the south, fuelled by the cold outflow from the Black Sea and by intense upwelling, could be the limiting factor preventing the invasion/establishment of tropical species [50], particularly of the ectothermic ones such as *P. segnis*. According to Zainal and Noorani [51] the thermal tolerance of *P. segnis* is between 5 °C and 40 °C, beyond which its survival becomes critical. Furthermore, a study on stress resistance of *P. pelagicus* larvae showed a very low survival rate at 10 °C and 20 °C but the highest survival was found at a constant temperature of 30 °C [52]. So, temperature may play a key role in shaping the colonisation, establishment and spread of *P. segnis* in the Mediterranean. Similar considerations may also apply to the attempts of this species to colonise the western Mediterranean basin that also seem to have been unsuccessful so far. The occurrence of larval and adult stages detected in the Ligurian Sea [32,53] have not given rise to an enlargement of the population but considering the slow times of expansion of this species we do not exclude the possibility in the future. 

### 4.2. *P. segnis* Invasion: Impacts on Biodiversity and Ecosystem Services

In accordance with our results *P. segnis* can be rightly categorized as an invasive alien species of Mediterranean Sea ecoregions (sensu Spalding et al. [54]): Levantine sea, Ionian Sea, and Tunisian plateau/Gulf of Sidra. The first tangible evidence of *P. segnis* invasiveness in the Mediterranean occurred when the species appeared on local markets as a fishing resource. In addition to the fishing markets of Haifa and Alexandria [16], this species was also sold in those of Syria and Turkey [25] shortly after, where it became commercially important particularly at local fish markets in Mersin and Iskenderun Bays [55]. A few decades later *P. segnis* also appeared in the fishing markets of eastern Sicily where a dedicated fishery began [28]. Here the species suffered a sharp decline in the early 1980s but then recovered [53]. More recently, it has acquired high commercial value and is increasingly in demand by consumers also in Tunisia [43] where it has established large populations and is becoming a value–added product for export to Asian and European markets [56]. On the one hand, *P. segnis* represents today a valuable fishery resource and then has a positive impact on food provision ecosystem services (*sensu* Liquete et al. 2013 [57]), on the other hand there are negative impacts on fishery economy that relate to the predation by blue swimming crabs on fishing catches and the damage of fishing nets where they get entangled [58,59]. Furthermore, given the continued expansion of the population of *P. segnis* in the Mediterranean, major negative impacts on biodiversity should be expected. In fact, this species is an opportunist predator, mainly preying on crustaceans, molluscs and fish [56] and then it is supposed to compete strongly and interfere with native species for resources and niches [59]. As an example, Ariani and Serra [28] reported the observations of fishermen on the rarefaction of two edible crabs, *Eriphia verrucosa* and *Carcinus aestuarii*, in conjunction with the explosion of *P. segnis* population in eastern Sicily, probably also due to the observed increased pollution in the area since the late 1950s, from industrial and petrochemical plants as well as from agriculture and urban waste [60], which may have made native crab populations more fragile. 

*P. segnis* has also had a positive impact on recreation and tourism as well as on cognitive effects (cultural ecosystem services) since it is an easily sightable species, reaches large dimensions, lives at shallow depth and has a characteristic attractive colour pattern. It is therefore an excellent target for lovers of snorkeling and SCUBA divers who are always looking for sightings of fascinating organisms. Various educational experiences carried out at primary schools or public information events (e.g., European Researchers’ Night) have shown blue crab to be the ideal organism to inform and raise public opinion on non–indigenous species phenomenon and their impacts. *P. segnis*, as well as other blue crab species, may also represent valid material for art applications. 

### 4.3. *P. segnis* Invasion: Management Actions 

Limiting the population expansion of *P. segnis*, or containing it where it is already settled, requires urgent management actions. Considering the 8Rs model (i.e., Recognize, Reduce, Replace, Reuse, Recycle, Recover/Restore, Remove, and Regulate) proposed by Rotter et al. [61], at least six of eight NIS mitigation strategies on this species can be undertaken. The blue crabs (*sensu lato*: i.e., also including *C. sapidus*) are easily *Recognizable* due to their typical features, so they are the ideal subject for the involvement of citizen science with the purpose of early detection and monitoring, as has already happened in the Pelagie Islands Marine Protected Area (Strait of Sicily) where an informative campaign on non–indigenous species carried out within a research project allowed the collection of the first and subsequent records of *P. segnis* in the area [8]. Citizens might also be involved in *Removal* strategies, e.g., through blue crab inclusion as target species in fishing tourism activities as well as the promotion of crab fishing competitions. Similar actions have already proven to be valid, as happened in Florida and Cyprus for the removal of the invasive lionfish with consequent policy changes, e.g., allowing lionfish fishing also in protected areas (*Regulate* rule) [62,63]. The blue crabs have very palatable and increasingly appreciated meats so that they reach high commercial value in local fish markets. A selective dedicated fishery, like that using trotlines to fish *Callinectes sapidus* in the U.S. [64], would then *Remove* and *Reuse* them as food and consequently *Reduce* their population, preserving the environment. Blue crab exploitation would also allow chitin and chitosans extraction (*Recycle* rule) for a variety of applications, mainly in the field of pharmaceutical and food industry [65,66], with a positive impact on provisioning ecosystem services. Finally, long–term monitoring specifically carried out on blue crabs could be considered within Conventions and Directives in order to *Regulate* control measures at national and international level. Among the potential mitigation measures to adopt, the exploitation and marketing of blue crab as a high–value fishery resource could be a feasible strategy to implement, as also suggested by [67] for the control of *C. sapidus* populations. 

## 5. Conclusions

The systematic review carried out on *Portunus segnis* on a broad geographical scale, such as the Suez Canal-Mediterranean Sea area, proved to facilitate the investigation of the geodistribution of an invasive alien species. Such a method allows retrieval of the major bulk of information, reducing the risk of losing data or of using invalid information. 

The identification of invasion pathways, main settlement and spread areas of invasive species by GIS analysis is crucial information for supporting decision makers to properly address management actions. This approach should be considered in particular when bioinvasions become a cross-border phenomenon since it allows early warning measures useful to prevent undesired impacts to be set up. 

The management of IAS in the marine environment still shows many gaps, even more so when it comes to species of commercial interest such as blue crabs, for which guidelines would be needed in support of decision makers at Mediterranean level. Continuous monitoring of blue crabs, together with more current early detection strategies such as the training of citizens as sentinels of the sea, would allow an improvement of knowledge as well as a saving in time and resources. Such actions would be particularly useful in identifying any new areas of settlement by blue crabs before significant environmental impacts can occur.

PS. As part of a citizen science campaign on alien species, in August 2022 the presence of *P. segnis* in the harbour of the Island of Pantelleria (Strait of Sicily), located between Tunisia and Sicily was documented by Giorgio Blandino and identified by the marine biologist Costanza Bonomo. This confirms the colonisation northwards of this species, as already suggested by our GIS analyses.

## Figures and Tables

**Figure 1 biology-11-01473-f001:**
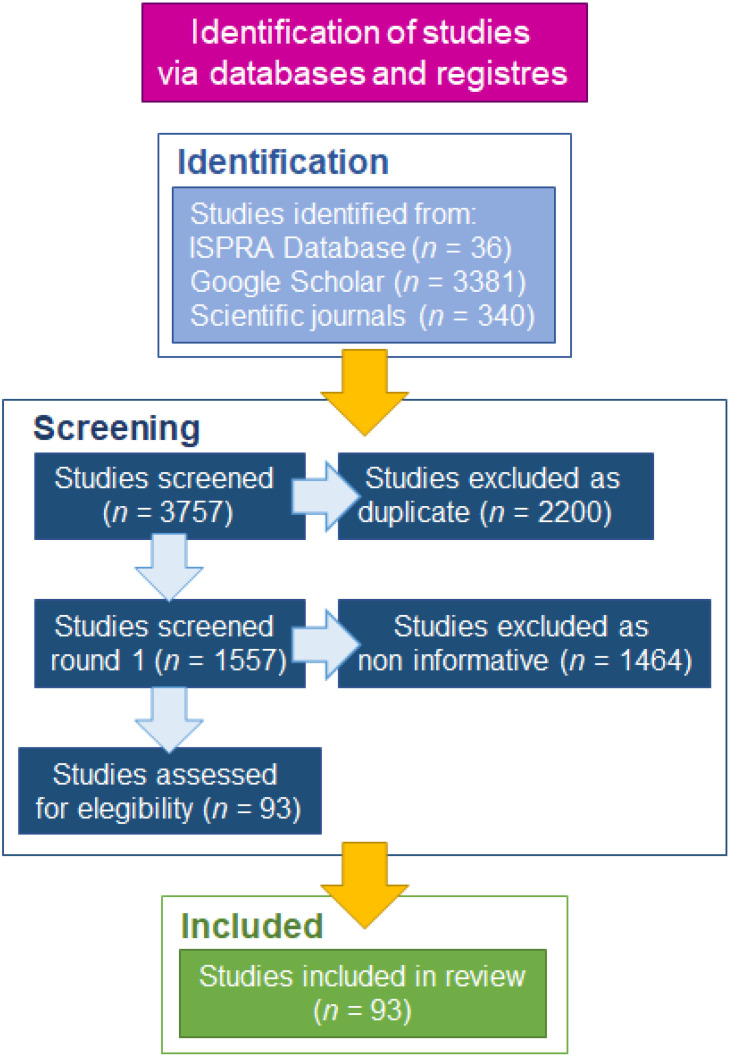
Flow chart presenting consecutive steps of the process of identification, screening, and eligibility of references for the systematic review.

**Figure 2 biology-11-01473-f002:**
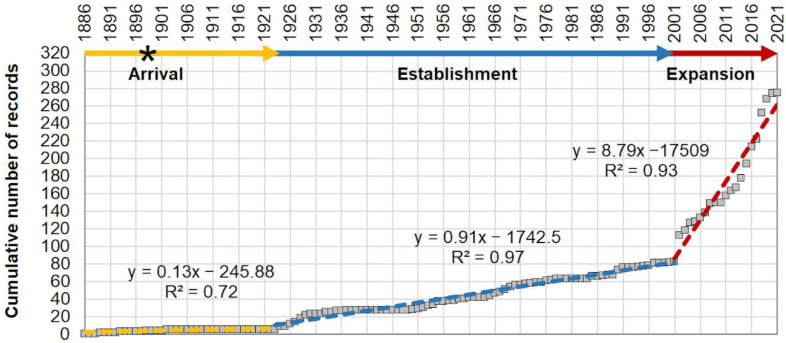
Cumulative curve of occurrences of *P. segnis* in the Mediterranean with indication of the three phases in the invasion process: arrival, establishment and expansion. The equations of the three regression lines with the correspondent R2 are also reported. The asterisk (*) indicates the first record in the Mediterranean Sea in 1898.

**Figure 3 biology-11-01473-f003:**
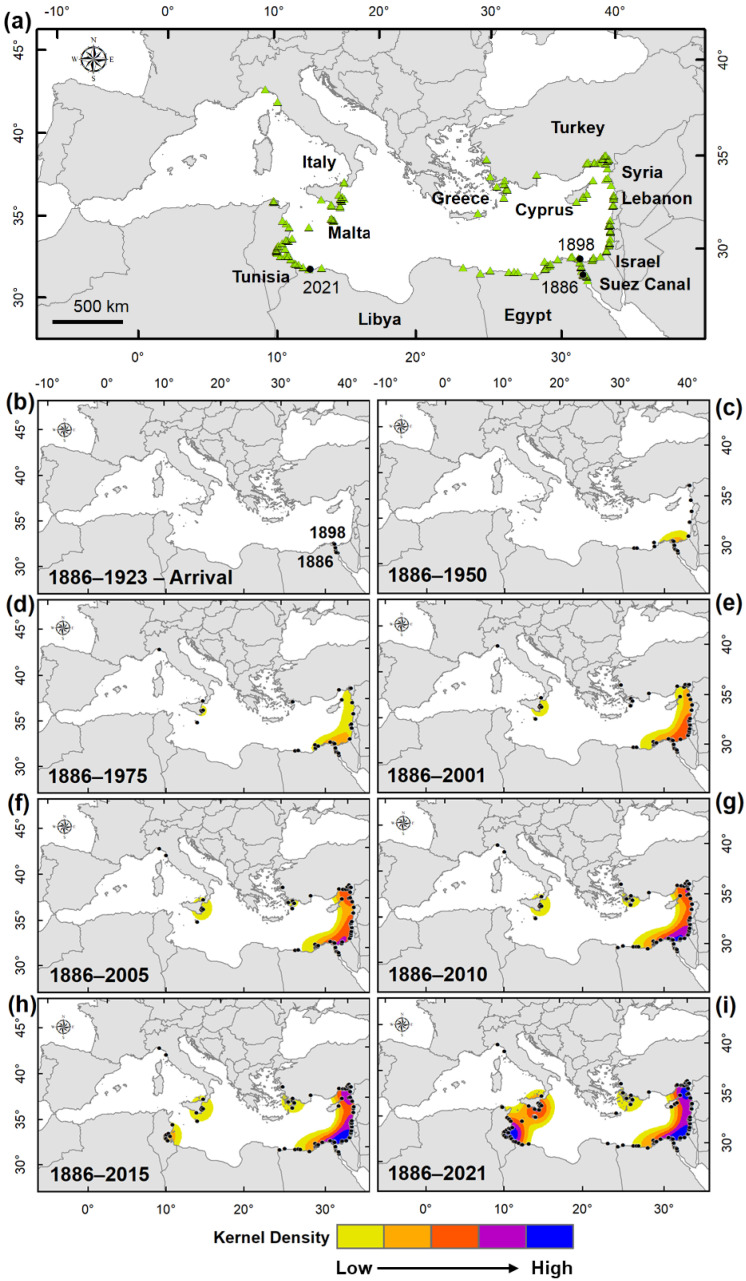
Distribution of *Portunus segnis* in the Mediterranean Sea together with the Kernel density cumulative maps. (**a**) The overall distribution of the selected records; (**b**–**i**) Period–to–period variation in space and time of *P. segnis* occurrence in Mediterranean Sea, (**b**) 1886–1923 period corresponding to the arrival phase of *P. segnis*. The first record in the Suez Canal in 1886 and the first record in the Mediterranean in 1898 are reported. (**c**–**e**) 1886–1950, 1886–1975, 1886–2001 periods. (**d**–**i**) 1886–2005, 1886–2010, 1886–2015, 1886–2021 periods. The triangles and the circles in (**a**–**i**) indicate the records of *P. segnis* in the Mediterranean Sea. First record in Suez Canal (1886) and in the Mediterranean Sea (1898), and the last record in Libya (2021) are also reported in (**a**) (circles).

**Figure 4 biology-11-01473-f004:**
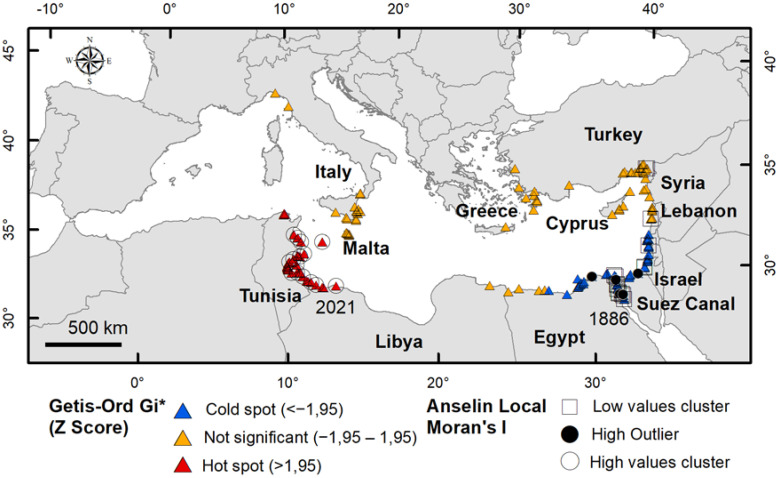
Results of the hot spot (Getis–Ord Gi*), and of the cluster and outlier analysis (Anselin local Moran’s I) on records of *Portunus segnis* in Mediterranean Sea. Areas with statistically significant spatial clustering (hot spot—red triangles and cold spot—blue triangle; cluster of high values—ring and cluster of low values—square) were identified and high outliers were detected. Yellow triangles indicate records with non–significant index values. The first (1886) and the last (2021) occurrences of *P. segnis* are also reported.

**Figure 5 biology-11-01473-f005:**
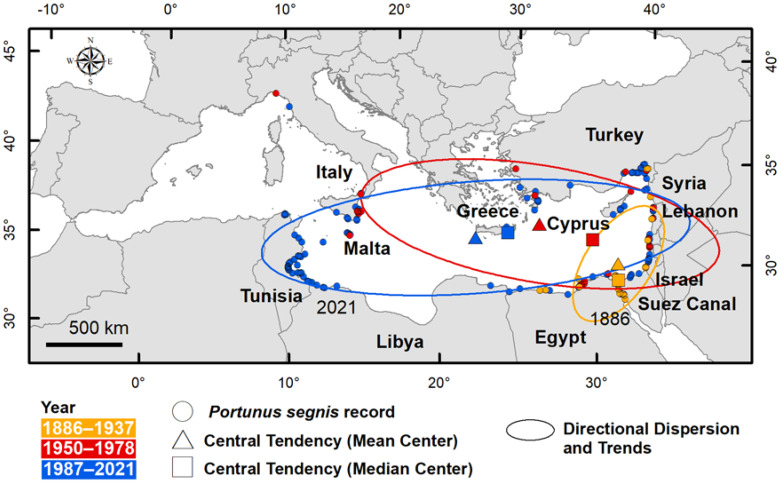
Distribution key characteristics of *Portunus segnis* in the Mediterranean Sea. The central tendency (such as mean and median centre), directional dispersion and trends, calculated for 1886–1937, 1950–1978 and 1987–2021 periods, show distribution changing in space and time. The first (1886) and the last (2021) occurrences of *P. segnis* are reported.

**Table 1 biology-11-01473-t001:** Spatial and temporal indicators and their ecological meaning, including methods and spatial and time scale used in the present study (modified from Perzia et al. [7]).

Analysis/Indicator Name	Tools	Spatial Scale	Time Scale	Ecological Meaning
**Temporal and Spatial–Temporal Pattern**				
Population increase	Cumulative curve of occurrence	Global	All years	Occurrences increasing over time
Population’s increasing rate	Evaluation of the slopes of the cumulative curve by Least Squares Method	Global	1886–1923 1924–2001 2002–2021	The rate of specimens’ increasing in time
Density hotspots	Kernel density	Global	1886–1923 1886–1950 1886–1975 1886–2001 1886–2005 1886–2010 1886–2015 1886–2021	Highest density areas; nuclei of record aggregation; occurrence persistent areas; space–time occurrences density increase.
**Aggregation patterns and spatial structure**				
Global Spatial Autocorrelation	Global Moran’s I (GMI) cutoff distance = 250 km	Global	All years	Distribution pattern: dispersion vs. random vs. clustering. Change in the spatial pattern over time
Statistically significant hot spots and cold spots	Getis—Ord Gi* (GOG*) Hot spot analysis cutoff distance = 250 km	Local	All years	Direction of spread and identification of dispersion/settle areas
Spatial outliers	Anselin local Moran’s I (AMI) Cluster and outlier analysis search threshold = 250 km	Local	All years	Direction of spread and identification of dispersion/settle areas and outliers
**Key characteristics of distribution**				
Centre of gravity	Central tendency (mean centre—median centre)	Global	1886–1937 1950–1978 1987–2021	Species concentration centre and its changing over time
Directional Dispersion	Standard deviational ellipse (1 standard deviation)	Global	1886–1937 1950–1978 1987–2021	Species distribution in X and Y directions
Directional Trends	Standard deviational ellipse (1 standard deviation)	Global	1886–1937 1950–1978 1987–2021	Directional trend of species dispersion

**Table 2 biology-11-01473-t002:** Values of the key characteristics of *Portunus segnis* records’ distribution in the Mediterranean Sea calculated per time period: central tendency (mean and median centre), directional dispersion and directional trends. DD = Decimal Degrees.

Indicator	Method	Unit	1886–1937	1950–1978	1987–2021
Central tendency	Mean centre	Longitude (DD) Latitude (DD)	32.62 31.98	27.46 35.13	22.97 34.79
Central tendency	Median centre	Longitude (DD) Latitude (DD)	32.39 31.06	31.33 33.58	25.75 34.93
Directional dispersion	Standard deviational ellipse	XStdDist (km) YStdDist (km)	219 429	366 1405	1178 381
Directional trends	Standard deviational ellipse	Rotation (°)	33	99	86

## Data Availability

No new data were created or analyzed in this study. Data sharing is not applicable to this article.

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
