# Peer review of "The Blue Swimming Crab *Portunus segnis* in the Mediterranean Sea: Invasion Paths, Impacts and Management Measures"

_biology, 2022, doi:10.3390/biology11101473_

Round 1

Reviewer 1 Report

In the paper "The blue swimming crab Portunus Segnis in the Mediterranean Sea: Invasion paths, impacts and management measures" the authors deal with the phenomenon of a potentially harmful invasive alien species on a broad geographical scale. The work is very interesting and above all it connects two processes often treated in a separate way: introduction and expansion. The work is well written and the bibliographic references are accurate. however, I believe that there is a lack of a formal component that can give strength to the work. The analysis of metadata and occurrences, made on bibliographic sources, require the support of classical methodologies, a workflow on data management, on the research, validation and exclusion process. Without a formal path this work is much less incisive than it could be. I recommend the authors to follow a procedure like the PRISMA: The PRISMA 2020 statement: an updated guideline for reporting systematic reviews. For the rest, the authors should insert in the paragraph "materials and methods" the part relating to the statistics that is reported in the results.

Author Response

Dear Reviewer, we revised the manuscript biology-1935160 according to your  comments. We report here our revision in attachment.

Reviewer 2 Report

Dear Authors,

The presented manuscript titled „The blue swimming crab Portunus segnis in the Mediterranean Sea: Invasion paths, impacts and management measures” contains very valuable results. Due to considerable scientific interest in problem of biological invasions I am convinced that submitted manuscript might attract atention of the large international audience. However, I have found some imperfections, which- in my opinion- should be improved or clarified before an eventual publication. Please, find them below:

1.       In my opinion the description of a study species in lines 77-88 is too scarce. Please, add information about biology of this crustacean (e.g. life span, diet, habitat etc.) and point out the traits, influencing on the substantial invasivness of  Portunus segnis.

2.       Lines 94-108. In my opinion the description of metod of systematic review of literature should be enlarged. Please,

-justify the use of chosen databases (why Web of Science was not used);

- add criteria of exclusion and inclusion of publications;

-add information about number of excluded and included papers.

Very valuable would be chart presenting consecutive steps of studies.

Perhaps below-listed publication would be helpful:

·         Moher D, Liberati A, Tetzlaff J, Altman DG; PRISMA Group. Preferred reporting items for systematic reviews and meta-analyses: the PRISMA statement. PLoS Med 21;6(7):e1000097. doi: 10.1371/journal.pmed.1000097

·         Page MJ, Moher D, Bossuyt PM, Boutron I, Hoffmann TC, Mulrow CD, Shamseer L, Tetzlaff JM, Akl EA, Brennan SE, Chou R, Glanville J, Grimshaw JM, Hróbjartsson A, Lalu MM, Li T, Loder EW, Mayo-Wilson E, McDonald S, McGuinness LA, Stewart LA, Thomas J, Tricco AC, Welch VA, Whiting P, McKenzie JE. PRISMA 2020 explanation and elaboration: updated guidance and exemplars for reporting systematic reviews. BMJ 2021;372:n160. https://www.bmj.com/content/372/bmj.n160

3.       The performed review of literature sources might be base to formulate the proposed directions of future investigations. Such information should be placed in chapter Conclusions.

Author Response

(The authors gave the same response as above.)

Reviewer 3 Report

This study analyses the distribution, aggregation patterns, and spatial structure of an invasive species in the Mediterranean Sea, the blue swimming crab Portunus segnis. The authors used ecological indicators with GIS spatial–temporal statistics to describe the different invasion phases. Furthermore, mitigation/management actions are proposed based on the 8Rs model. This manuscript addresses an important issue in the study of invasion pathways and processes in the Mediterranean Sea using GIS tools, and it will be of great interest to readers of Biology.

The manuscript is generally well-written, the introduction includes and summarize all important information and reference, methods are clearly stated. However, punctuation and the use of too many long sentences can be improved.

Minor revisions are briefly reported hereafter:

Line 54: Avoid too long sentences

Line 102: “recognises” instead of “recognise”

Line 104: “Data were” instead of “was”

Line 112: Please explain better what you mean by “source”.

Line 116: Please add punctuation and brackets to improve clarity, for example: 1. exact coordinates or detailed map reported on the document; 2. indication of the specific sighting/capture site (e.g., city name); 3. indication of the generic sighting/capture site (e.g., gulf name); 4. Highly generic location (e.g., sea name).

Line 244: Please briefly explain the “key characteristics”

Line 277“through their inclusion instead of “through its inclusion”

Conclusion: This section just includes comments on “Citizen Science”, please add comments about outcome results/discussion from the GIS analyses.

Author Response

(The authors gave the same response as above.)

Round 2

Reviewer 2 Report

Dear Authors,

In my opinion presented manuscript received sufficient improvement therefore I do not have any further suggestions of changes.